

# Artificial intelligence system can achieve comparable results to experts for bone age assessment of Chinese children with abnormal growth and development

Fengdan Wang[1,*], Xiao Gu[2,*], Shi Chen[2], Yongliang Liu[3], Qing Shen[3], Hui Pan[2], Lei Shi[3] and Zhengyu Jin[1]

[1] Department of Radiology, Peking Union Medical College Hospital, Chinese Academy of Medical Sciences & Peking Union Medical College, Beijing, China
[2] Department of Endocrinology, Peking Union Medical College Hospital, Chinese Academy of Medical Sciences & Peking Union Medical College, Beijing, China
[3] Hangzhou YITU Healthcare Technology Co., Ltd., Hangzhou, China
[*] These authors contributed equally to this work.

Corresponding authors
Hui Pan, panhui20111111@126.com
Lei Shi, lei.shi@yitu-inc.com

## ABSTRACT

**Objective**. Bone age (BA) is a crucial indicator for revealing the growth and development of children. This study tested the performance of a fully automated artificial intelligence (AI) system for BA assessment of Chinese children with abnormal growth and development.

**Materials and Methods**. A fully automated AI system based on the Greulich and Pyle (GP) method was developed for Chinese children by using 8,000 BA radiographs from five medical centers nationwide in China. Then, a total of 745 cases (360 boys and 385 girls) with abnormal growth and development from another tertiary medical center of north China were consecutively collected between January and October 2018 to test the system. The reference standard was defined as the result interpreted by two experienced reviewers (a radiologist with 10 years and an endocrinologist with 15 years of experience in BA reading) through consensus using the GP atlas. BA accuracy within 1 year, root mean square error (RMSE), mean absolute difference (MAD), and 95% limits of agreement according to the Bland-Altman plot were statistically calculated.

**Results**. For Chinese pediatric patients with abnormal growth and development, the accuracy of this new automated AI system within 1 year was 84.60% as compared to the reference standard, with the highest percentage of 89.45% in the 12- to 18-year group. The RMSE, MAD, and 95% limits of agreement of the AI system were 0.76 years, 0.58 years, and −1.547 to 1.428, respectively, according to the Bland-Altman plot. The largest difference between the AI and experts' BA result was noted for patients of short stature with bone deformities, severe osteomalacia, or different rates of maturation of the carpals and phalanges.

**Conclusions**. The developed automated AI system could achieve comparable BA results to experienced reviewers for Chinese children with abnormal growth and development.

## INTRODUCTION

Bone age (BA), which evaluates skeletal maturity from the radiographs of the left hand and wrist, is a crucial indicator for revealing the growth and development of children (*Creo & Schwenk, 2017*). Two methods are mainly used to assess BA: the Greulich and Pyle (GP) and Tanner-Whitehouse (TW3) (*Greulich & Pyle, 1959*; *Tanner, 1962*; *Tanner et al., 1983*; *Tanner et al., 2001*); of these, the GP atlas is generally accepted as a faster and simpler method and thus widely applied in clinical practice (*De Sanctis et al., 2014*). However, manual assessment of BA completely depends on the reviewers' experience to determine BA (*Creo & Schwenk, 2017*; *Mari, 2015*), thereby causing significant intra- and inter-observer variations. Furthermore, constant time and effort are needed to train clinical reviewers; consequently, primary and rural hospitals face a daunting task to carry out this important examination.

Artificial intelligence (AI), which has high potential in reducing labor requirement and intra- and inter-observer variations, is gaining popularity in medical field, especially in radiology (*Cheng et al., 2019*; *Hosny et al., 2018*). Deep learning, one of the advanced AI techniques, which can automatically learn features from images, has become a hot spot in recent years. BA images are an ideal database to train deep learning algorithms, because bone radiographs contain black-white-gray gradations that show variations (*Hu et al., 2017*).

In addition to some traditional learning-based approaches (*Van Rijn & Thodberg, 2013*; *Thodberg et al., 2017*), several preliminary deep learning-based BA systems have been developed using a standard database or radiographs from one or two medical centers in North America and Korea (*Kim et al., 2017*; *Spampinato et al., 2017*; *Larson et al., 2018*; *Tajmir et al., 2019*; *Halabi et al., 2019*). However, these deep learning-based systems were developed for Western and Korean populations; thus, they might not be suitable for Chinese children. Moreover, the number of tested patients was relatively small and only with limited types of diseases. Therefore, these AI BA systems cannot be directly applied to populations of different ethnicities, and a fully automated AI system for BA assessment of Chinese children should hence be developed. In addition, the test sample should be larger and should include patients with a wide range of diseases, because children with abnormal growth and development are more likely to have impaired bone maturation and malformations that pose greater challenge for AI.

Herein, a fully automated AI BA system for Chinese children based on the GP method was developed using 8000 BA radiographs from five medical centers nationwide in China (X Zhou et al., 2019, unpublished data). This study aimed to evaluate the performance of this AI system for BA assessment of Chinese children with endocrine disorders. The test cases used to validate this AI system were patients with abnormal growth and development and were consecutively recruited from another tertiary medical center in north China.
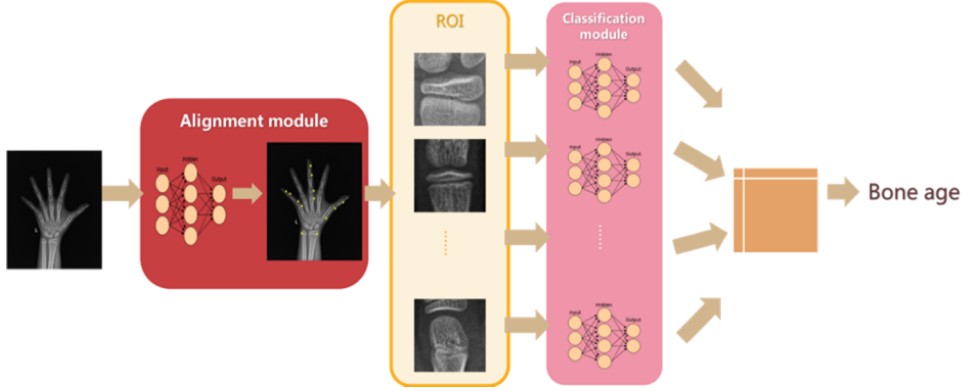

**Figure 1** **Network structure of the region-based convolutional neural network.** The left hand radiograph is preprocessed, and the alignment module then adopts a deep residual network (ResNet) to localize the bounding area of the targeted bones. Subsequently, the classification module also adopts a ResNet for extracting features of targeted bones and provides bone age result.

## SUBJECTS AND METHODS

### Model implementation

This study was approved by the Institutional Review Board of Peking Union Medical College Hospital (Approval No. S-K978), and informed consent was waived because this was a retrospective study. The patients were anonymized by de-identifying their personal information prior to analysis. Between January 2012 and December 2016, 8,000 left-hand radiographs from five different medical centers nationwide in China were used as input data for developing the AI model (X Zhou et al., 2019, unpublished data). Our medical center is not one of the five medical centers that provided the radiographs. The demographic characteristics of the 8,000 cases (training dataset of 7,000 cases, validation dataset of 760 cases, and test dataset of 240 cases) are shown in (Fig. S1 & Table S1). The BA results extracted from the original radiology reports were used as the reference standard.

The AI model consisted of an alignment module and a subsequent classification module (Fig. 1). The two modules were built on the same architecture, known as a deep residual network (ResNet), which is a deep convolutional neural network (CNN) with 50 layers and approximately $3.6 \times 10^9$ floating point operations (FLOPS). The model was implemented using an open-source machine learning library (TensorFlow version 1.4.1; Google, Mountain View, CA, USA). The model was trained on an Ubuntu 16.04 computer with 14 Intel Xeon CPUs, using a NVIDIA GTX 1080 Ti 11 Gb GPU, with 256 GB random access memory (RAM).

Before training, each radiograph was first converted from DICOM to portable network graphic (PNG) file format. The radiographs contained images of distal ulna, distal radius, carpal, and metacarpal and phalangeal bones with a resolution of at least $1,000 \times 1,000$ pixels formatted using Python (version 3.7) and the pydicom library (Python Software Foundation; version 0.9.9, Beaverton, OR, USA). The images were further downsized to $256 \times 256$ pixels by using Python image library.

After pre-processing of the clinical radiographs, the alignment module adopts a ResNet to directly regress all the coordinates of 59 localized points. These characteristics are used for determining the bounding area of the concerned bones and ossification center. The region of interest (ROI) can then be extracted from hand radiographs, and the classification module can extract features from the concerned bones and ignore the region unnecessary for BA assessment.

Training of the model was performed by stochastic gradient descent in batches of 20 images per step by using an Adam Optimizer with a learning rate of 0.001. Training on all categories was run for 80,000 iterations, because training of the final layers will have converged by then for all classes. After 80,000 iterations through the entire dataset, the training was stopped due to the absence of further improvement in both accuracy and sigmoid loss.

### Test patient population

From January to October 2018, left-hand BA radiographs were consecutively collected from 753 pediatric patients aged 4 to 18 years who presented to our medical center with a complaint of abnormal growth and development. Except for this criterion, we chose patients without any age and sex preference. Among the 753 patients, radiographs of eight patients were excluded because of poor imaging quality. Finally, a total of 745 radiographs (male to female ratio = 1:1.07) were included to test the AI system. The flowchart of case collection is shown in Fig. 2.

The de-identified DICOM images were downloaded from the PACS system and then inputted into the AI system for batch processing. The AI system automatically generated the BA results.

### Reference standard

The reference standard was provided by two trained and experienced reviewers (a radiologist with 10 years and an endocrinologist with 15 years of experience in BA reading) using the GP atlas. Both reviewers assessed all the 745 cases through consensus. They were blinded to patient information, diagnosis, treatment, and previous BA reports apart from sex and age. There was no time limit to assess all radiographs. In the case of a disagreement, a third reviewer, an endocrinologist specialized in child growth and development with over 20 years of experiences in BA reading, was consulted.

Regarding the GP atlas, "Skeletal development of the hand and wrist—a radiographic atlas and digital BA companion" published by Oxford University Press in 2011 was adopted. This digital atlas has been widely used in clinical practice as an efficient and accurate method (*Bunch et al., 2017*).

### Statistical analysis

To compare the evaluation results of the AI system and human reviewers, several statistical variants were used. BA accuracy was defined as the percentage of the differences between the two methods within 1 year. Pearson's correlation coefficient was calculated to analyze the relativity. Bland-Altman plots were used to calculate the mean and 95% confidence interval of the difference between them. Root mean square error (RMSE) and median

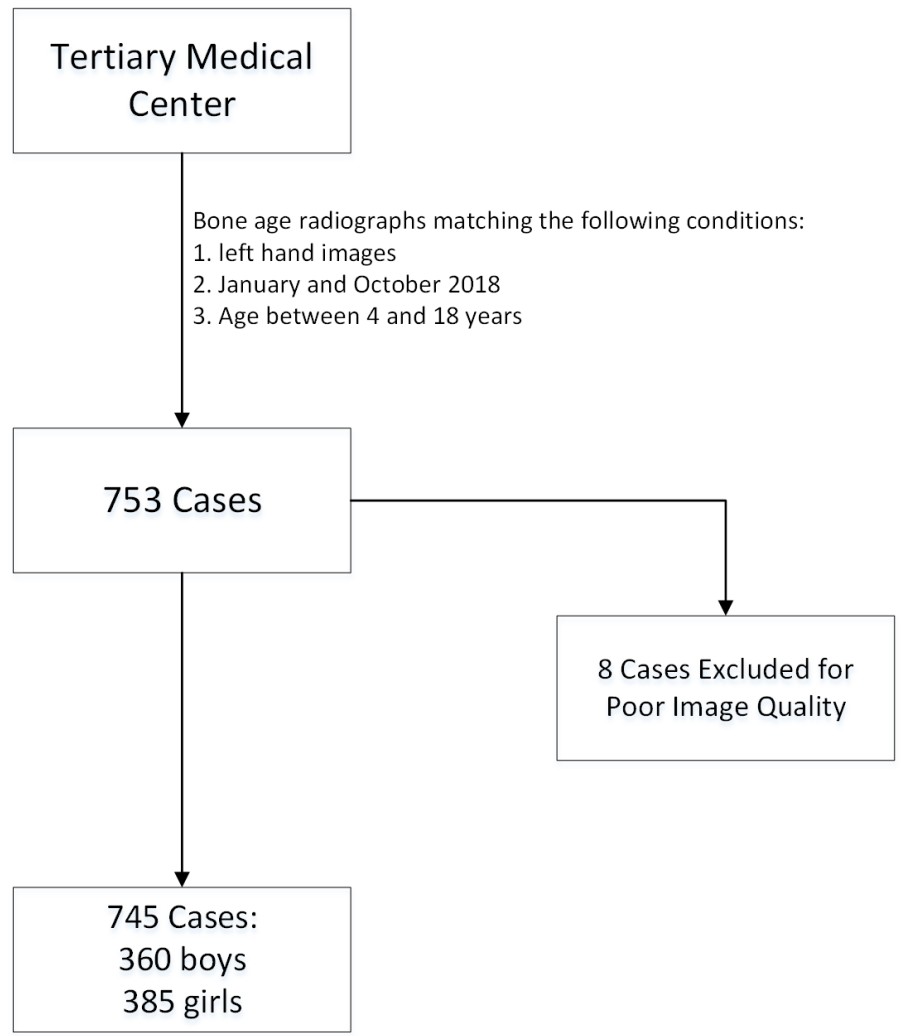

**Figure 2** The flowchart of cohort selection with inclusion and exclusion criteria.

absolute deviation (MAD) were quantified to evaluate the accuracy. Statistical differences were considered to be significant at $p < 0.05$. Calculations were performed using SPSS v.22.0 software (SPSS Inc., Chicago, IL, USA), and Bland-Altman plots were created using GraphPad Prism 7 statistical software (Graphpad Software, San Diego, CA, USA).

## RESULTS

### Test patient population

Between January and October 2018, a total of 745 BA radiographs were included to test the AI BA software. All radiographs were obtained from Chinese children with abnormal growth and development. The demographic information of the 745 patients is presented in Table 1. There were 360 males and 385 females, with a mean age of 10.2 years (range: 4–18 years). Most patients (420/745, 56.38%) were included in the 6- to 12-year group,

**Table 1  The demographic characteristics of the 745 Chinese patients.**

| | Characteristics | Number (%) | |
|---|---|---|---|
| **Gender** | Male | 360 (48.32%) | |
| | Female | 385 (51.58%) | |
| **Age (years)** | 4 | 21 (2.82%) | 126 (16.91%) |
| | 5 | 59 (7.92%) | |
| | 6 | 46 (6.16%) | |
| | 7 | 53 (7.11%) | |
| | 8 | 46 (6.17%) | |
| | 9 | 118 (15.84%) | 420 (56.38%) |
| | 10 | 68 (9.13%) | |
| | 11 | 70 (9.40%) | |
| | 12 | 65 (8.72%) | |
| | 13 | 43 (5.77%) | |
| | 14 | 78 (10.47) | |
| | 15 | 32 (4.30%) | 199 (26.71%) |
| | 16 | 26 (3.49%) | |
| | 17 | 13 (1.74%) | |
| | 18 | 7 (0.94%) | |
| **Diagnosis** | Growth hormone deficiency | 214(28.72%) | |
| | Turner syndrome | 81(10.87%) | |
| | Precocious puberty | 77(10.34%) | |
| | Congenital adrenal hyperplasia | 72(9.66%) | |
| | Hypogonadism | 25(3.36%) | |
| | Small for gestational age | 22(2.95%) | |
| | Kallmann's syndrome | 15(2.01%) | |
| | Hypothyroidism | 13(1.74%) | |
| | Hypospadias | 10(1.34%) | |
| | Mixed gonadal dysgenesis | 9(1.21%) | |
| | Noonan syndrome | 6(0.81%) | |
| | Prader-willi syndrome | 5(0.67%) | |
| | Cryptorchidism | 5(0.67%) | |
| | Systemic juvenile idiopathic arthritis | 5(0.67%) | |
| | Renal tubular acidosis | 5(0.67%) | |
| | McCune-Albright syndrome | 3(0.40%) | |
| | Pseudohypoparathyroidism | 2(0.27%) | |
| | Gynecomastia | 2(0.27%) | |
| | Gigantism | 1(0.13%) | |
| | Short stature for unknown reasons | 173(23.22%) | |

followed by 199 patients (26.71%) in the 12- to 18-year group and 126 patients (16.91%) in the 4–6-year-old group. The main reasons for presentation to the clinic were short stature, precocious puberty, macrosomia, and congenital disorders.

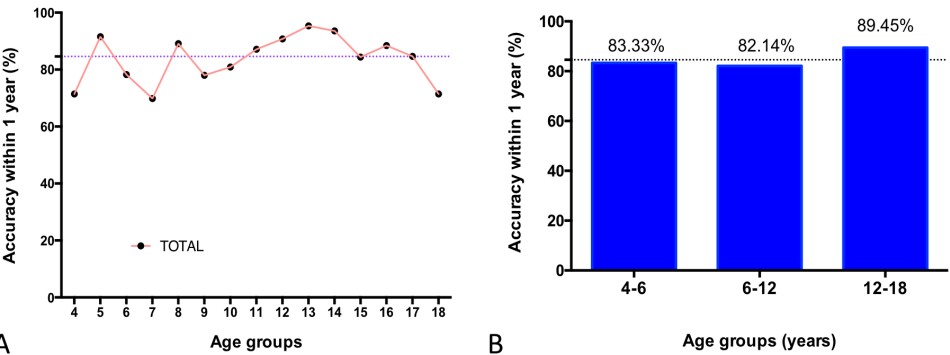

**Figure 3  The accuracy of AI BA determined by comparing with the reference standard for different age groups.** Distribution of BA accuracy within 1 year is shown according to chronological age (A) and three age groups (4–6, 6–12, and 12–18 years) (B).

## Accuracy of the automated AI BA system for Chinese children

For reading one radiograph, the two experienced reviewers took approximately 2 min on average, while the AI model required only 1 to 2 s. Consequently, the two reviewers spent over 24 h analyzing the 745 cases, while the AI system batch processed all the images together generating the output in an excel in less than 1 h; this showed that the AI system was significant efficient compared to manual analysis.

By using the two experienced reviewers' manual reading results as a reference standard, the overall BA accuracy of AI within 1 year was 84.60%. The distribution of BA accuracy of AI within 1 year is illustrated in Fig. 3, and one example of a case is shown in Fig. 4A. If we categorized the results into the three age groups (4–6, 6–12, and 12–18 years), the highest percentage of BA accuracy within 1 year was 89.45% in the 12- to 18-year.

In addition, the agreement of the AI BA results with the reference standard was further quantified using RMSE, MAD, and the Bland-Altman plot. The RMSE of AI was 0.76 years and the MAD was 0.58 years (95% confidence interval, 0.55 to 0.62 years) when compared with the manual reference standard. The 95% limits of agreement of the AI system and the reference standard was −1.547 to 1.428 according to the Bland-Altman plot (Fig. 5).

## Analysis of the largest deviation

Five cases with a deviation of ≥ 2 years were noted between the AI and manual BA results. To further understand the confounders that caused this deviation, their clinical features and BA radiographs were further analyzed; the results are shown in Table 2 and Figs. 4B to 4D.

For patient No. 1, the deformity of the left fifth middle phalange was identified. For patient No. 2, the first inter-phalangeal joint curved and thus obscured the fusion line of the distal phalangeal joint; the epiphysis of the first distal phalangeal joint was the only one with fusion in this boy, which served as an important benchmark to determine the BA according to the GP atlas. In these two cases, the reviewers could identify the deformity and malposition easily, while the AI system could not.

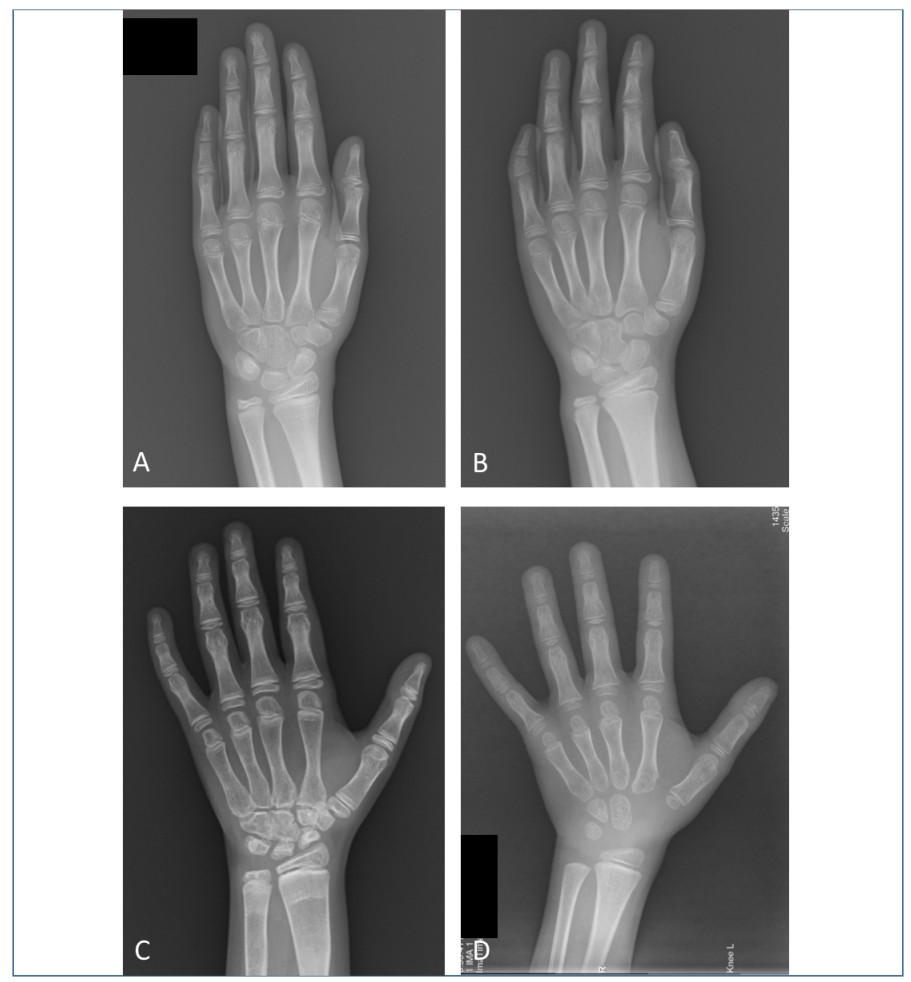

**Figure 4   Examples of left hand bone age (BA) radiographs from four different patients.** (A) Left hand radiograph from a 9.3-year-old girl with precocious puberty. Both the manual and AI BA result is 10 years. (B) Left hand radiograph from an 8.2-year-old girl with idiopathic short stature. The manual BA result is 8.8 years, and the AI BA result is 11.1 years. The fifth middle phalange is short and without a normal epiphysis. (C) Left hand radiograph from a 13.4-year-old girl with systemic juvenile idiopathic arthritis and receiving long-term treatment with corticosteroids. The manual BA result is 9 years, and the AI BA result is 11.7 years. There is severe osteomalacia of hand and wrist bones, and the shapes of the carpal bones are irregular. (D) Left hand radiograph from a 7.3-year-old boy with Turner syndrome. The manual BA result is 4.5 years, and the AI BA result is 2.1 years. The different rates of maturation of the carpals and phalanges may cause this deviation in the results.

Patient No. 3 received long-term treatment with corticosteroids for systemic juvenile idiopathic arthritis, which led to severe osteomalacia of hand and wrist bones and irregular shapes of the carpal bones. For patients Nos. 4 and 5, the rates of maturation of the carpals and phalanges were different. In accordance with the tips of the GP atlas, the phalanges were given more priority than carpal bones by the reviewers. Therefore, achieving precise BA results for these three cases was difficult for both the experienced reviewers and the AI system.

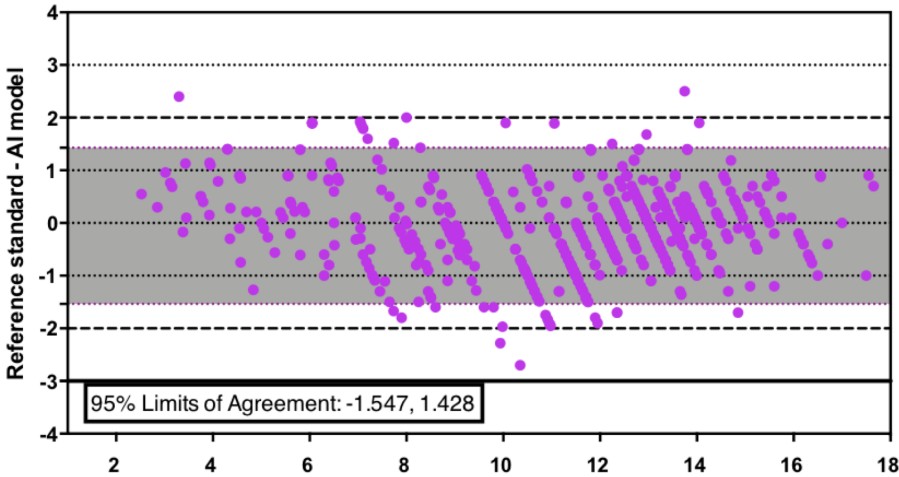

**Bland-Altman**

95% Limits of Agreement: -1.547, 1.428

Mean of AI model and the reference standard of BA by radiologist (years)

**Figure 5** The Bland-Altman plot showing the difference between the AI BA and the reference standard.

**Table 2** Clinical features of the five cases with the largest BA deviation (≥2 years) between AI and the reference standard.

| Patient number | Sex | Chronological age (years) | Diagnosis | Bone age (years) | |
|---|---|---|---|---|---|
| | | | | Reviewers | AI |
| 1 | F | 8.2 | Idiopathic short stature | 8.8 | 11.1 |
| 2 | M | 13.4 | Growth hormone deficiency | 15 | 12.5 |
| 3 | F | 13.4 | Systemic juvenile idiopathic arthritis | 9 | 11.7 |
| 4 | M | 10.5 | Idiopathic short stature | 9 | 7 |
| 5 | M | 7.3 | Turner syndrome | 4.5 | 2.1 |

If these five cases were excluded, the overall BA accuracy of the AI system within 1 year was elevated to 85.10%, the RMSE decreased to 0.73 years, and the MAD was 0.57 years (95% confidence interval, 0.54 to 0.61 years). Additionally, the 95% limits of agreement of the AI system and the reference standard was −1.494 to 1.379 according to the Bland-Altman plot.

## DISCUSSION

In this study, we trained a fully automated AI system to assess BA of Chinese children by using 8,000 radiographs from five different medical centers nationwide in China. We then evaluated the performance of the AI system by using 745 radiographs of patients with abnormal growth and development from another medical center. Compared to the interpretation results of experienced human reviewers, the overall BA accuracy of AI

within 1 year was nearly 85%. According to the results of RMSE, MAD, and 95% limits of agreement, the degree of dispersion was also acceptable, indicating that this AI system performs well as a reliable and convenient tool for BA assessment of Chinese children in terms of saving time and energy.

AI systems based on deep learning have a bright future for BA assessment clinically. Some computer-aided programs of BA estimation have been proposed, including analyzing bones using the GP method (*Hsieh et al., 2007*) and establishing an algorithm based on the TW3 method (*Liu et al., 2008*; *Van Rijn & Thodberg, 2013*). However, only in recent years, with the development of deep learning, the research on BA assessment has entered a new era, and several AI systems have been developed to assess BA in North America and Korea. BoneXpert, a traditional machine learning commercial system, is based on a feature extraction technique that reconstructs the borders of the bones. It is reported that the MAD between this system and manual assessments ranged from 0.55 to 0.76 years (*Van Rijn & Thodberg, 2013*). Nevertheless, not all the bones in one BA radiograph are considered, and the system cannot identify pathological conditions such as malformations (*Kim et al., 2017*). In 2017, Kim et al. developed an AI BA system that provided three most likely estimated BA results for one radiograph based on cases from the Asan Medical Center and then tested it on 200 cases evenly distributed by age from the same medical center; the authors obtained a first rank accuracy of 69.5% (*Kim et al., 2017*). In 2018, 200 sex-stratified cases from Stanford University, one of the two American medical centers where Larson et al. trained their AI BA system, were used to test its accuracy; the system achieved an RMSE of 0.63 years (*Larson et al., 2018*). In 2019, Tajmir et al. tested their AI BA system using 280 cases comprising 10 representative cases for each class and sex and chosen from 8,325 radiographs which they used to train the system; the BA accuracy was 73.2% (*Tajmir et al., 2019*). Herein, we tested our fully automated AI BA model with radiographs from 745 Chinese pediatric patients with abnormal growth and development. This number of test patients (745 cases) was almost triple of that used in previous studies (*Kim et al., 2017*; *Larson et al., 2018*; *Booz et al., 2019*; *Tajmir et al., 2019*) and also the largest one for Chinese population. The results showed that this AI BA system for Chinese children exhibited similar accuracy to the experienced reviewers.

Prior studies of BA assessment based on deep learning usually trained and tested the program with radiographs from the same database with similar epidemiological characteristics (*Kim et al., 2017*; *Larson et al., 2018*; *Tajmir et al., 2019*). However, real clinical circumstances are more complicated, as the AI program is supposed to be applied in various medical centers facing patients with a wide range of diseases. Our training data set was chosen from five different medical centers in five different cities nationwide in China, and the test cases were from a sixth medical center in north China. The use of radiographs from different places for algorithm development and program evaluation could simulate the clinical application of the AI program. Moreover, our cohort was consecutively selected without age or sex preference, which was more like a mimic of clinical situation.

Of note, the performance of our algorithm was best for children aged between 12 to 18 years. Skeleton development is more mature for the older children in the 12–18 years age group because all the 29 bones of the hand have developed. For younger children, the

appearance of hand bones varies; thus, there were less ROIs as input for extracting features and BA assessment in the radiographs of younger children. Consequently, the accuracy of AI BA assessment for children younger than 12 years may be less accurate than that for children aged 12–18 years. In clinical practice, BA assessment for children younger than 12 years is also more difficult for radiologists and endocrinologists, and interobserver variability is more likely to occur (*Ebrahimzade et al., 2019*; *Alshamrani, Messina & Offiah, 2019*). Therefore, more training data of left hand radiographs from younger children are helpful to improve the performance of the AI BA system.

The phalanges and metacarpal, and carpal bones grew and developed differently in two cases of the 5 patients with a deviation of over 2 years between the AI system and the reference standard; this made the hand radiograph resemble several standard GP images obtained at different ages, thus causing the difficulty in BA assessment for both AI and experienced human readers (*Molinari, Gasser & Largo, 2004*; *Zhang et al., 2008*). This unsynchronized bone growth may result from certain diseases or abnormal secretion of hormones (*Kim et al., 2010*; *Polito et al., 1994*). Moreover, radius, ulna, and short bones are formed by enchondroplasia, while carpal bones are formed by chondral osteogenesis, which is less dependent on growth hormones (*Mari, 2015*). When using the GP atlas, the phalanges and metacarpals are more emphasized (*Larson et al., 2018*). Nevertheless, even when images with such large deviations were included in the testing set, the overall accuracy of our AI system was nearly 85%.

It has remained controversial whether AI would replace radiologists. Similar to the good performance of AI in detecting skin cancer (*Esteva et al., 2017*), diabetic retinopathy (*Gulshan et al., 2016*), and breast cancer (*Rodríguez-Ruiz et al., 2019*), various AI systems (*Larson et al., 2018*; *Bui, Lee & Shin, 2019*; *Liu et al., 2019*) were reported to achieve comparable BA results to those of experienced readers. With the advantage of artificial neural networks and deep learning, AI can "see" (abstract) subtle imaging characteristics to learn and improve the algorithm automatically, thus overcoming the limitations of time, energy, expenses, and intra- and inter-observer variations of human radiologists. However, AI algorithms for image recognition must be developed on the basis of "labeled data" where BA results interpreted by human radiologists are taken as the ground truth (*Davenport & Dreyer, 2018*). Therefore, researchers are now heading toward the model in which both AI and radiologists can work cooperatively rather than competitively (*Siegel, 2019*). It has been proven that readers with the assistance of AI can achieve better BA accuracy than readers alone or AI alone (*Kim et al., 2017*; *Tajmir et al., 2019*). In our study, radiologists could identify deformities and malposition from hand radiographs concurrently while giving the BA results, but these are technical obstacles for AI processing. In the future, a more precise BA workflow is expected that integrates AI with radiologists' practice.

Our research still had some limitations. First, the test sample size was still relatively small for an AI validation, but our number of patients was much larger than that of most prior studies and the largest one among Chinese population. Second, we did not investigate whether the AI system could help human reviewers to improve the accuracy of BA assessment, as the AI system was more likely to become an auxiliary method for human reviewers. Third, we did not test the performance of our algorithm on the RSNA dataset

because our algorithm was developed using a different dataset with quite different imaging quality and demographic information. Lastly, more efforts should be made to improve the accuracy of this AI BA system and to broaden its age range.

## CONCLUSION

The developed automated AI system could achieve comparable BA results to experienced reviewers for Chinese children with abnormal growth and development.

## ACKNOWLEDGEMENTS

The authors thank Bo Liu (Hangzhou YITU Healthcare Technology Co., Ltd.) for his technical support and Dr. Huadan Xue (Department of Radiology, Peking Union Medical College Hospital, Chinese Academy of Medical Sciences & Peking Union Medical College) and Dr. Daming Zhang (Department of Radiology, Peking Union Medical College Hospital, Chinese Academy of Medical Sciences & Peking Union Medical College) for their help during proposal design.

### Funding

This study was supported by the Chinese National Public Welfare Basic Scientific Research Program of Chinese Academy of Medical Sciences (Grant No. 2018PT32003 and 2017PT32004), and the Beijing Municipal Natural Science Foundation (Grant No. 7192153). The funders had no role in study design, data collection and analysis, decision to publish, or preparation of the manuscript.

### Grant Disclosures

The following grant information was disclosed by the authors:
Chinese National Public Welfare Basic Scientific Research Program of Chinese Academy of Medical Sciences: 2018PT32003, 2017PT32004.
Beijing Municipal Natural Science Foundation: 7192153.

### Competing Interests

Yongliang Liu, Qing Shen and Lei Shi are employees of Hangzhou YITU Healthcare Technology Co., Ltd., in charge of developing the AI model. The other authors declare no financial relationships with Hangzhou YITU Healthcare Technology Co., Ltd.

### Author Contributions

- Fengdan Wang, Xiao Gu and Yongliang Liu conceived and designed the experiments, performed the experiments, analyzed the data, prepared figures and/or tables, authored or reviewed drafts of the paper, and approved the final draft.
- Shi Chen conceived and designed the experiments, performed the experiments, prepared figures and/or tables, authored or reviewed drafts of the paper, and approved the final draft.

- Qing Shen performed the experiments, analyzed the data, prepared figures and/or tables, authored or reviewed drafts of the paper, and approved the final draft.
- Hui Pan and Lei Shi conceived and designed the experiments, authored or reviewed drafts of the paper, and approved the final draft.
- Zhengyu Jin conceived and designed the experiments, authored or reviewed drafts of the paper, foresee the future of AI in radiology and guide the whole team to realize it, and approved the final draft.

### Human Ethics

The following information was supplied relating to ethical approvals (i.e., approving body and any reference numbers):

The Institutional Review Board of Peking Union Medical College Hospital approved this study (S-K978).

### Data Availability

The raw measurements of chronological age and bone age, including the 8,000 cases used to develop the AI system and the 745 cases used to test the system are available as Supplementary Files.

### Supplemental Information

Supplemental information for this article can be found online at http://dx.doi.org/10.7717/peerj.8854#supplemental-information.

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
