# Peer review of "Artificial intelligence system can achieve comparable results to experts for bone age assessment of Chinese children with abnormal growth and development"

_PeerJ, doi:10.7717/peerj.8854_

## Round 0.1 · original submission · Major Revisions

You are welcome to revise your manuscript based on the comments, and re-submit it to PeerJ

Reviewer 1 ·

Basic reporting

This work describes a study using deep learning to predict bone age from X-ray images of the left hand, which is widely used in endocrinological assessment of delayed or accelerated growth in children. A database of 8000 Chinese children from 5 different Chinese clinics is used to train the deep learning model. It consists of X-ray images and the clinically reported bone age. For testing, a database of 745 further children from a different Chinese clinic is used, where bone age is assessed as consensus of two raters. The deep learning method is compared with this ground truth, leading to a mean absolute deviation of 0.58 years and an RMSE of 0.76 years.

The paper is reasonably well written and provides a good overall structure. However, a main concern is the rather superficial literature survey, which misses many relevant works in bone age assessment. Most importantly, the result of the recent RSNA bone age estimation challenge (doi: 10.1148/radiol.2018180736), which is based on the dataset of Larson et al., 2018, is entirely ignored in this work. However, at that challenge, a number of state of the art deep learning algorithms were discussed, many of them very similar to the method the authors propose in this work (and describe as a "new" fully automatic method). It is unclear how the proposed method relates to the methods presented there. There is also the work of Spampinato et al. in Medical Image Analysis and the extensive work of Thodberg et al. and van Rijn et al., which is entirely ignored in this paper, although these can be considered state of the art methods in bone age estimation, both for deep learning based and more traditional learning based approaches. Given that this work presumably is not in its first iteration as a journal submission, I see this neglectance of related work (indicated by the sentence in line 70: "To date, a few preliminary AI BA systems have been developed based on a standard database or radiographs from one or two medical centers in North America and Korea, with a test sample size of 200.", which in my opinion is very unfair given the huge amount of work Hans Henrik Thodberg has already invested into validating his commercially available BoneXpert methdod!) and the ignoring of the wider context of automated bone age estimation and its in the meantime quite extensive literature as hugely problematic, when assessing the quality of this work.

The claimed contributions of this work are firstly, that a new fully automated AI system for BA assessment is developed (stated throughout the manuscript), and secondly, that it is dedicated to Chinese children, since growth and development may differ from those ethnicities that were used for other previously developed machine learning approaches. While I agree that the second contribution is very relevant - the ethnical differences are indeed an interesting topic to study - the first issue is not a contribution for two reasons. 1. The proposed method using a Residual Net for localization of age relevant structures and a Residual Net for age prediction is not novel. 2. It is not clear why we need yet another proposed deep learning method for bone age assessment from X ray images, given that there are so many around already (see above, especially those from the RSNA challenge), and that the authors do not indicate, what distinguishes their method from the state of the art, and how their method performs on the publicly available datasets that can be used for comparison. Therefore, I do not think that the hypotheses stated in this work are demonstrated through a convincing experimental setup (see also 2. Experimental design).

Experimental design

There are a number of detailed concerns that I have regarding this work:

- Ethically, I would prefer that the ethics committee of the university approves the use of the data for this retrospective study. The currently uploaded document is not convincing me that the ethical prerequisites are fulfilled. Just because the dataset is retrospective and anonymized does not mean that the ethical committee should not still approve this study. We have received this approval by our ethics committee for our studies, and also e.g. Larson et al., Radiology 2018 describe that use of their retrospective dataset was approved by the local ethics committee.

- In this work there is no description of the characteristics of the training dataset used for the deep learning approach. It is solely stated that 8000 images from five centers were used. No information on the age distribution, how the reference standard was obtained, the sex characteristics or the diagnostic information (as in Table 1 for the test dataset) is available. Therefore it is hard to judge if the dataset was appropriate for training. Also no information on the hyperparameters, their tuning and the use of a validation set for this tuning is given. Since we have to trust that no test image was ever used during the development of the proposed deep learning algorithm, it would be more reassuring to see on which data the hyperparameter tuning was performed. Otherwise one has to assume that hyperparameter tuning was done on the test set, thus leading to results that would poorly generalize to future new datasets.

- The reference standard section describes the generation of the test set's reference standard as the consensus of two reviewers who were unblinded from sex and age information. The knowledge of the chronological age, however, is not valid for creating this reference standard, since the aim of bone age assessment is exactly to determine the difference to chronological age (which would be the average age of "normal" development) in order to assess endocrinological disorders. Also Larson et al., 2018 make it very clear that their reference standard was developed blinded to chronological age information. I see this as a severe limitation of the generated reference standard, and additionally, the design of the crucial reference standard is lacking compared to the effort put into this difficult step e.g. by Larson et al., 2018 and Halabi et al., Radiology 2019 (4 assessors, one additional assessment after some time, correction for rater bias).

- It is entirely unclear how the landmark localization pre-processing depicted in Fig. 1 works. This is a crucial step, since errors in this step will have a severe impact on the age estimation error. A very vague description of the use of residual networks for landmark localization is given, however, this strategy is not a state of the art method in landmark localization, where recently heatmap regression approaches and U Net like structures are seeing a lot of attention. Furthermore, the landmark localization approach is difficult to implement given the wide age range of children that have to be assessed (6-18 years). This is mentioned in the manuscript, but not discussed as to how it impacts the landmark localization!

- The discussion of the limitations of the proposed method is not very extensive, however, it would add important insights of the study outcome.

Validity of the findings

The proposed results are not properly embedded into the related work, especially not compared on the publicly available databases to give the context and to justify the reason why a new fully automatic algorithm has to be developed, instead of using an existing one that was already shown to be suitable for bone age assessment. This makes it hard to compare the proposed method on the new dataset of Chinese children, and it makes it hard to judge the specific properties of the newly collected dataset. In addition to this, the Bland Altman plot with its very large deviation in the form of the 95% limits of agreement does not convince me that the proposed method is a clinically suitable bone age predictor, despite the claims made in this work. Presumably, this error is in practice too large to justify automatic processing. However, given that the reference standard was determined by "approximately two minutes on average" of investigating the X ray image (see line 168), this prediction error may also come from the reference standard itself, as mentioned in section 2 (experimental design). This makes it hard to judge the result on the challenging test dataset that is used in this study.

Overall, while I applaud the collection of the large database of training and testing images, and the effort put into validating deep learning based age estimation on a different ethnicity, I do not think that the presented study is thorough enough to justify publication. Since it is not possible to reproduce results due to the lack of training and testing database, the data annotation and the code of the proposed method, the quality of the experiments supporting the claimed hypothesis have to be of a very high standard to be convincing. I do not see this important prerequisite demonstrated in the presented study.

Reviewer 2 ·

Basic reporting

The authors present a good and interesting article on application of soft computing techniques to predict the bone age of Chinese children with abnormal growth and development. This topic is relevant considering the importance of elimination of time-consuming consultation necessary to analysis.

The manuscript demonstrates that the authors understand relevant literatures enough in the field of application of soft computing algorithms to investigate growth and development of children. The existing literature is up-to-date and it is well cited (However, I recommend to replace citations before the "." in the text, for instance, line 66, 69, 143, 223, 226, 229, 237, 257, 264, 266, 268, 269 ...).

In general, the used statistical analysis and artificial intelligence-based models are suitable, and the presentation of the results is very clear. The structure of this paper and the quality of figures are very good (I just recommend to modify in Figure 2 from yeas to years).

The raw data are correctly shared.

Therefore, I recommend this paper for publication in the journal after solving some minor issues previously mentioned.

Experimental design

This research is very interesting and falling within the scope of the journal. The experiments are well designed. The database is well collected and preprocessed, providing sufficient information. The age of patient is well distributed from 4 to 14 years. The experiments providing by two experienced reviewers are very good, allowing independent confrontations for the AI model. The materials and methods are described with sufficient detail and information to replicate by other researchers.

Therefore, I highly appreciate the materials and methods section of this paper.

Validity of the findings

The results are statistically analyzed. The discussion section is well given, including limitations of this work. The authors also attempt to control extreme cases, this is very good. The results of the AI model is compared with manual investigations from independent investigators, proving the relevant contribution of this work in reducing labor requirement.

The proposed methodology of this paper is very useful. I recommend the authors to improve the accuracy of the AI model and supply it to the field of Health Sciences as an additional robust tools.

---

## Round 0.2 · Minor Revisions

Please address the reviewer's concerns and check through your paper for any potential grammar mistakes.

Reviewer 1 ·

Basic reporting

Compared with the previous version of the paper, where the literature review was insufficient, the manuscript has made a big step in terms of being embedded in the current literature. The overall quality of the paper is now suitable for publication.

Experimental design

The authors have done a very good job in rebuttling the criticisms regarding experimental design. The issues on clarity of the training setup, the algorithm used for predictions, and the concerns regarding the performance in the context of the related work, as well as the doubts about the reference standard were clarified. The original criticism regarding the knowledge of chronological age for constructing the reference standard and for the limited time spent with an image to create the Greulich Pyle based reading were probably too harsh, and I have reconsidered these through reading the rebuttal.

There is still one issue that I would like to mention, after reading the new manuscript and the rebuttal. It requires some refined discussion in a potential final revision:

- The use of regions of interest extracted before performing age prediction solely on those regions is not novel, and is in my opinion not strong enough to support a granted patent. While this idea has been pioneered by Tanner and Whitehouse already in an attempt to improve the traditional atlas method of Greulich and Pyle, it was later also used by BoneXpert through their statistical shape model approach or by Pietka et al., 2004 in J Digital Imag. Very recently this idea was also demonstrated in deep learning based age estimation from MRI by Stern et al., IEEE JBHI 2019 or Stern et al., Medical Image Analysis 2019, where anatomical landmarks are localized using heatmap regression (see Payer et al., Medical Image Analysis 2019).

Actually, I think the novelty claim of the proposed method is probably not very important for this manuscript, since it should focus on the study of the Chinese population, including the separate dataset for final thorough validation and generalization. This is the strongest point of this work. Nevertheless, I find it important that the methodological state-of-the-art is reflected in its entirety.

Validity of the findings

Overall, compared to the previous revision, I am now convinced that the findings are valid, and very relevant to be reported.

Additional comments

There are a few grammar mistakes that were introduced in the revised version. I would suggest another proofread.

---

## Round 0.3 · accepted · Accept

Thanks for your submission.